# Weight Pulling: A Novel Mouse Model of Human Progressive Resistance Exercise

**DOI:** 10.3390/cells10092459

**Published:** 2021-09-17

**Authors:** Wenyuan G. Zhu, Jamie E. Hibbert, Kuan Hung Lin, Nathaniel D. Steinert, Jake L. Lemens, Kent W. Jorgenson, Sarah M. Newman, Dudley W. Lamming, Troy A. Hornberger

**Affiliations:** 1Department of Comparative Biosciences, University of Wisconsin-Madison, Madison, WI 53706, USA; wzhu89@wisc.edu (W.G.Z.); jhibbert@wisc.edu (J.E.H.); klin52@wisc.edu (K.H.L.); nsteinert@wisc.edu (N.D.S.); jlemens@wisc.edu (J.L.L.); kwjorgenson@wisc.edu (K.W.J.); 2School of Veterinary Medicine, University of Wisconsin-Madison, Madison, WI 53706, USA; 3Department of Medicine, University of Wisconsin-Madison, Madison, WI 53705, USA; snewman2@wisc.edu (S.M.N.); lamming@wisc.edu (D.W.L.); 4William S. Middleton Memorial Veterans Hospital, Madison, WI 53705, USA

**Keywords:** skeletal muscle, hypertrophy, myonuclear accretion, protein synthesis, signaling, growth, strength, mTOR

## Abstract

This study describes a mouse model of progressive resistance exercise that utilizes a full-body/multi-joint exercise (weight pulling) along with a training protocol that mimics a traditional human paradigm (three training sessions per week, ~8–12 repetitions per set, 2 min of rest between sets, approximately two maximal-intensity sets per session, last set taken to failure, and a progressive increase in loading that is based on the individual’s performance). We demonstrate that weight pulling can induce an increase in the mass of numerous muscles throughout the body. The relative increase in muscle mass is similar to what has been observed in human studies, and is associated with the same type of long-term adaptations that occur in humans (e.g., fiber hypertrophy, myonuclear accretion, and, in some instances, a fast-to-slow transition in Type II fiber composition). Moreover, we demonstrate that weight pulling can induce the same type of acute responses that are thought to drive these long-term adaptations (e.g., the activation of signaling through mTORC1 and the induction of protein synthesis at 1 h post-exercise). Collectively, the results of this study indicate that weight pulling can serve as a highly translatable mouse model of progressive resistance exercise.

## 1. Introduction

Skeletal muscles contribute to ~40% of total body mass, and, aside from being the motors that drive basic movements, also play a critical role in breathing, whole-body metabolism, and maintaining a high quality of life [1,2]. Indeed, the average individual will lose 30–40% of their muscle mass by the age of 80, and this loss in muscle mass (i.e., sarcopenia) is associated with disability, loss of independence, and an increased risk of morbidity and mortality, as well as an estimated USD 18.5 billion in annual healthcare costs in the United States alone [3,4,5]. Accordingly, the development of therapies that can restore, maintain, and/or increase muscle mass will be of great clinical and fiscal significance. However, to develop such therapies, we will first need to establish a comprehensive understanding of the mechanisms that regulate the mass of this vital tissue.

Over the last century, it has become apparent that an increase in muscle mass can be induced by a variety of different stimuli, with one of the most widely recognized being progressive resistance exercise (PRE) [6,7]. For instance, numerous studies have shown that 8–16 weeks of PRE can lead to a 5–20% increase in skeletal muscle mass/volume, along with an even greater increase in strength [8,9,10]. The PRE-induced increase in muscle mass is also associated with improvements in markers of mental, metabolic, and cardiovascular health, as well as a reduction in body fat and the risk of developing/dying from aging-related diseases [11,12,13,14]. In other words, the health-related benefits of PRE go well beyond an increase in muscularity, yet the mechanisms that drive these effects remain far from defined.

During the last 50 years, a variety of different animal models have been used to develop a better understanding of how PRE induces an increase in muscle mass. The majority of these models have involved the use of rodents, with earlier models largely relying on rats and more recent models relying on mice [15,16,17,18,19]. Mouse models are now viewed as being particularly advantageous because they are amenable to the wide array of genetic inventions that are often paramount to mechanistic studies. With this point in mind, Murach et al. (2020) recently summarized the currently available mouse models and, as highlighted in their review, most models use chronic forms of mechanical overload (e.g., the synergist ablation model) and/or only focus on one or two muscles within a single limb [20]. Indeed, we are not aware of any mouse models that have been shown to induce an increase in the mass of numerous muscles throughout the body. This is an important point because the most robust health-related benefits of PRE would be expected to come from interventions that impact all of the major muscle groups. Thus, one of the primary goals of this study was to develop a mouse model that mimics a human PRE training paradigm and induces an increase in the mass of numerous muscles throughout the body. 

In order to develop a model that accurately mimics a human PRE training paradigm, we referred to the American College of Sports Medicine (ACSM) recommendations for progression models of resistance exercise [6]. Specifically, we focused on the recommendations for training paradigms that are intended to induce muscular growth in novice human subjects (i.e., untrained individuals with no prior resistance exercise experience). For such individuals, the ACSM recommends that all major muscle groups be trained two to three times per week, with an emphasis placed on multi-joint exercises that involve 8–12 repetitions per set, 1–2 min of rest between sets, and a total of one to three maximal-intensity sets of each exercise per training session. Moreover, the resistance employed during each training session should be progressively increased as the individual’s strength improves. Over the last decade, these recommendations have been widely accepted, and, therefore, served as the foundation for the training paradigm that was implemented in our mouse model. 

In addition to inducing an increase in muscle mass, we also wanted to develop a mouse model that could elicit some of the classic types of responses that are known to occur when humans engage in PRE. For instance, numerous studies have shown that a bout of PRE leads to an acute increase in the rate of protein synthesis and the activation of signaling through growth regulatory molecules, such as the mechanistic target of rapamycin complex 1 (mTORC1) [21,22]. It has also been shown that the PRE-induced increase in muscle mass can be attributed to an increase in the cross-sectional area of muscle fibers (i.e., hypertrophy), and/or an increase in the length of muscle fascicles/fibers [23,24]. Moreover, these changes are often accompanied by an increase in the number of myonuclei per fiber, as well as a fast-to-slow transition in the composition of the Type II fibers [25,26]. Accordingly, we used a variety of different assays to determine whether these types of responses were elicited by our model, which we refer to as “weight pulling”. 

## 2. Materials and Methods

### 2.1. Animals and Ethical Approval

All animal experiments were approved by the Institutional Animal Care and Use Committee of the University of Wisconsin-Madison (#V005375) or the William S. Middleton Memorial Veterans Hospital (Assurance ID: D16-00403). Male C57BL/6J (Jackson Laboratories, Bar Harbor, MA, USA) mice at 9–10 weeks of age were randomly assigned to one of the experimental interventions described below, given food (5001 Rodent Laboratory Chow, Purinal Mills, Gray Summit, MO, USA) and water ad libitum (unless otherwise indicated), and kept in a room that was maintained at 25 °C with a 12–h light cycle (lights off at 6:00 p.m.–6:00 a.m.). During terminal collections, the mice were anesthetized with 1–5% isoflurane mixed in oxygen and euthanized by cervical dislocation under anesthesia.

### 2.2. Weight Pulling Model Components

The custom-built components of the weight pulling model are described in Figure 1 and included a 120 cm long weight pulling track that was constructed out of impermeable plastic lumber (BestPlus, Somerville, MA, USA) and lined with grip support material (Con-Tact Brand #05F-C6B0B-06, Pomona, CA, USA). The custom-built components also included a weight pulling cart (Figure 1B,C). The base of this cart was built from the parts that were contained within a toy car kit (eBay, sold by Sktflyer, item #370838782830, China), and these parts allowed for the construction of a 6.8 cm wide × 8.0 cm long wheelbase. An 11.4 cm long section of an 80-well microfuge tube rack was then cut and secured to the top of the wheelbase with strength molding tape (3M, #03609, Saint Paul, MN, USA). Superglue was also used to attach a 10 cm long piece of 1.5 mm thick nylon string to the front corners of the wheelbase. Combined, these components were defined as the “unweighted cart” and weighed a total of 70 g (Figure 1B). To construct the “weighted cart”, strength molding tape was used to secure microfuge tubes to the bottom corners of an acrylic box (5 cm wide × 12.7 cm long × 10.2 cm tall, shopPOPdisplays, #CS001, Little Falls, NJ, USA) and the tubes were placed in a position that aligned with the wells of the microfuge tube rack. The weighted cart was then formed by placing these tubes into the wells of the unweighted cart and, as needed, the box was filled with various amounts of lead weight (Eagle Claw, Denver, CO, USA) (Figure 1C).

### 2.3. Training Paradigm

#### 2.3.1. Acclimation

All mice were provided with one acclimation session. During this session, the unweighted cart was attached to the mouse’s tail by interweaving a 10 cm long strip of vinyl tape (3M, #80610833842, Saint Paul, MN, USA) around the tail and the nylon string of the cart (Figure 1D). With the unweighted cart attached to the tail, the mice were placed at the start line on the weight pulling track and then familiarized with pulling the cart to the end of the 50 cm long weight pulling lane (Figure 1A). At the end of the lane, the mice encountered a resting house and, upon reaching the resting house, they were given 1 min to recover before being returned to the start line on the weight pulling track. Initially, the mice were motivated to pull the unweighted cart by touching their rear fur close to the lumbar vertebrae region. If the mice failed to make forward progress after being touched 3 times (with a 1 s interval between each touch), then an additional incentive was provided by delivering a 1 mA shock to the lumbar vertebrae region/tail (Precision Animal Shocker, Coulbourn Instruments, #H13–15, Lehigh Valley, PA, USA). The familiarization process was repeated a minimum of 8 times, and until the mouse voluntarily traversed the entire length of the weight pulling lane three consecutive times.

#### 2.3.2. Weighted Pulling Paradigm

The weighted pulling (WP) training paradigm was initiated two days after the completion of the acclimation session. The first training session in this paradigm was aimed at determining the mouse’s maximal pulling load and consisted of ~10–12 sets with 2 min of rest between each set. During this procedure, the cart was attached to the tail as described above, and the mice were motivated to pull the cart by touching their rear fur. If the mice did not make forward progress after being touched 3 times (with a 3 s interval between each touch), an additional 1 mA shock was delivered. The first set during this session consisted of pulling the unweighted cart and then the next three sets consisted of pulling the weighted cart with a total load of 300, 600, and then 750 g. After completing these sets, an additional 50 g was added to the weighted cart and the mouse completed another set. The process of adding 50 g to the weighted cart was repeated until a load was reached during which the mouse failed to traverse the entire length of the weight pulling lane. Failure during this session was defined as the inability of the mouse to make forward progress after 3 touches and 1 shock. Upon reaching failure, assistance (i.e., gentle pushing of the cart in a manner that would be analogous to a spotter helping a human complete a final repetition during resistance exercise) was provided so that the mouse could traverse the remaining length of weight pulling lane without the need for a further touch/shock incentive. At this point, the training session was complete, and the highest load successfully pulled along the entire length of the lane was defined as the mouse’s maximal pulling load. 

Three days after the first training session, the mice initiated the remainder of the WP training paradigm. The training sessions during this period were performed three times per week for a total of 36 sessions (i.e., on Monday, Wednesday, Friday, or Tuesday, Thursday, Saturday). External motivation during these sessions consisted of touching the rear fur as described above, and if the mice did not make forward progress after 5 touches (with a 3 s interval between each touch), an additional 1 mA shock was delivered. A total of 2 min of rest was given between each set, and each training session began with a warm-up set that only involved pulling the unweighted cart. The warm-up set was then followed by sets with loads representing 50, 75, 85, 90, 95, and 100% of the mouse’s previous maximal pulling load, respectively. During all subsequent sets, an additional 15 g was added to the weighted cart, and this cycle was repeated until a load was reached during which the mouse failed to traverse the entire length of the weight pulling lane. Failure during this portion of the training paradigm was defined as the inability of the mouse to make forward progress after 5 consecutive touches followed by 1 shock and then 2 additional touches. Upon reaching failure, assistance (as described above) was provided so that the mouse could traverse the remaining length of the weight pulling lane without the need for further touch/shock incentive.

#### 2.3.3. Control (Unweighted) Paradigm

To appropriately control for the stresses that are incurred during the WP paradigm (e.g., taping of the cart to the tail, touches, shocks, etc.), the control mice performed a total of 37 control training sessions. Each control training session was performed at the same time as a WP training session (6:00–9:00 p.m.) and each control mouse was always paired with a WP mouse. The control session began by attaching an unweighted cart to the tail (as described above), and then the mouse performed the same number of sets as the paired WP mouse, but additional weight was never added to the cart. Moreover, before each set, the control mouse was placed on a smooth surface and subjected to the same number of touches/shocks that were experienced by the paired WP mouse during each of its respective sets.

### 2.4. Rotarod, Body Composition, and Isometric Grip Strength

Prior to the animal’s final training session, rotarod performance (a marker of motor coordination) was assessed with Rotamex 5 System (Columbus Instruments, Columbus, OH, USA). To assess performance, an acclimation session was performed 1 day before the actual testing session and consisted of three familiarization trials that were separated by 15 min of rest. Each familiarization trial lasted 1 min and used preprogrammed settings of 0.3 RPM (increment speed), 10 (increment seconds), 5 RPM (start speed), and 40 RPM (end speed). At 24 h after the acclimation session, rotarod performance was measured during three trials that were separated by 15 min of rest. During each trial, the system was set at 1 RPM (increment speed), 8 (increment seconds), 5 RPM (start speed), and 40 RPM (end speed). The average time to fall (i.e., latency) during all three trials was determined and used as the marker of rotarod performance. 

At 48 h after the final training session, body composition was determined in awake mice using an EchoMRI body composition analyzer. Once completed, isometric grip strength was assessed with a dual-range force sensor (Vernier, Beaverton, OR, USA) that was connected to a custom-built horizontal platform (6 cm wide × 12 cm long) that had been covered with the same grip support material that was used to line the base of the weight pulling lane (Con-Tact Brand #05F-C6B0B-06, Pomona, CA, USA). During the grip strength measurements, the mice were first allowed to grasp the matting with all four paws. Then, while keeping the torso parallel with the platform, the mouse’s tail was slowly pulled away from the force sensor until the mouse could no longer maintain its grip. The highest force generated was recorded and the procedure was repeated 3 times with 5 min of rest between each recording. The highest force measured during all three recordings was recorded as the animal’s isometric grip strength. Importantly, all of the measurements described in this section were performed by blinded investigators.

### 2.5. Terminal Collections after 13-Weeks of Training

Terminal collections were performed 60–96 h after the final training session. During this procedure, all potentially identifiable information (e.g., tail markings, etc.) was masked, and then the animals were weighed and subsequently anesthetized with isoflurane. The extensor digitorum longus (EDL), flexor digitorum longus (FDL), and soleus (SOL) muscles from both the left and right hindlimb were then weighed, submerged in optimal cutting temperature compound (OCT, Tissue-Tek; Sakura Finetek, The Netherlands) at resting length, and frozen in liquid nitrogen-chilled isopentane. At this point, the mice were euthanized by cervical dislocation, a photograph that included both a scale bar and the musculature of the left forelimb was obtained, and then additional skeletal muscles from both the left and right side of the body were collected, including the: tibialis anterior (TA); gastrocnemius (GAST); plantaris (PLT); tibialis posterior (TP); quadriceps (QUAD); semitendinosus (ST); teres major (TM); pectoralis major (PEC); lateral head of the triceps brachii (Tri-Lat); long head of the triceps brachii (Tri-Long); short head of the biceps brachii (Bi-Short); and the forearm flexor complex (FF), which consisted of the flexor carpi ulnaris, flexor carpi radialis, flexor digitorum superficialis, and all three heads of the flexor digitorum profundus. In addition to skeletal muscles, tissues including the heart, as well as the left and right tibias, epididymal fat pads, and adrenal glands were also collected post-mortem. All of the collection procedures were performed by blinded investigators.

### 2.6. Immunohistochemistry

Mid-belly cross-sections (10 μm thick) from muscles frozen in OCT were taken with a cryostat and fixed for 10 min at room temperature with 1% paraformaldehyde dissolved in PBS (for myonuclei staining) or for 10 min with −20 °C acetone (for fiber type staining). Fixed sections were washed with PBS for 15 min and then blocked for 20 min at room temp in buffer A (0.5% Triton X-100, 0.5% BSA dissolved in PBS). 

Myonuclei staining was performed as previously described [27]. Briefly, blocked samples were incubated for 1 h at room temp in buffer A containing mouse anti-dystrophin IgG1 (1:20, #NCL-DYS2, Novocastra/Leica, Buffalo Grove, IL, USA). The sections were then washed 3 times for 15 min with PBS and incubated in buffer A containing Alexa 488 goat anti-mouse IgG1 (1:500, #115-545-205, Jackson Immunoresearch, West Grove, PA, USA) for 1 h at room temp. Sections were then washed with PBS 3 times for 15 min and incubated with PBS containing Hoechst (1:5000, #561908, BD Pharmingen, Lincoln Park, NJ, USA) for 5 min before a final series of washes with PBS.

For fiber type staining, blocked samples were subjected to a 1 h incubation at room temp in buffer A containing rabbit anti-laminin (1:500, #L9393, MilliporeSigma, Burlington, MA, USA), mouse IgG2b anti-Type I MHC (1:100, # BA-D5-s, Developmental Studies Hybridoma Bank, Iowa City, IA, USA), mouse IgG1 anti-Type IIA MHC (1:100, #SC-71-s, Developmental Studies Hybridoma Bank, Iowa City, IA, USA), and mouse IgM anti-Type IIB MHC (1:10, #BF-F3-s, Developmental Studies Hybridoma Bank, Iowa City, IA, USA). The sections were then washed 3 times for 15 min with PBS and incubated for 1 h at room temp in buffer A containing Alexa 568 goat anti-rabbit IgG (1:5000, #A11011, Invitrogen, Waltham, MA, USA), Alexa 647 goat anti-mouse IgG2b (1:100, #115-605-207, Jackson Immunoresearch, West Grove, PA, USA), Alexa 488 goat anti-mouse IgG1 (1:3000, #115-545-205, Jackson Immunoresearch, West Grove, PA, USA), and Alexa 350 goat anti-mouse IgM (1:500, #A-31552, Invitrogen, Waltham, MA, USA). The sections were then subjected to a final series of washes with PBS. All washed sections were mounted in a layer of ProLong Gold anti-fade mounting medium (Invitrogen, Waltham, MA, USA), overlaid with a coverslip (Thermo Fisher Scientific, Waltham, MA, USA), and then whole muscle cross-sections were imaged by a blinded investigator with a 10X objective on a BZ-X700 Keyence microscope and 4 different filters (DAPI, GFP, TRITC, CY5) (Keyence, Itasca, IL, USA).

### 2.7. Image Analyses

For fiber cross-sectional area measurements, images of the whole muscle cross-section were analyzed and the cross-sectional areas (CSA) of all qualified fibers of each fiber type (Type I, IIA, IIB, and non-Type I, -Type IIA, or -Type IIB, which were defined as Type IIX) were counted and measured using our fiber cross-sectional area CellProfiler pipeline, which was based on modifications of the Muscle2View pipeline [28] (see Appendix A for more details). For myonuclei and interstitial nuclei identification, our myonuclei CellProfiler pipeline that was based on modifications of the Muscle2View pipeline was employed [28]. In this case, the whole muscle cross-section was analyzed and the average fiber CSA, total fiber number, total number of myonuclei, as well as total nuclei were determined (see Appendix A for more details). The distinction between myonuclei and interstitial nuclei was achieved by defining nuclei that resided within the dystrophin layer of individual myofibers as myonuclei, whereas those that resided outside of the dystrophin layer were designated as interstitial nuclei.

### 2.8. Acute Changes in Protein Phosphorylation and Protein Synthesis 

The rate of protein synthesis and the phosphorylation state of various proteins were assessed in the FDL muscles from a subset of the mice. Specifically, all of the mice in this subset performed the acclimation procedure and 3 sessions of training with WP. Prior to the fourth and final training session, the mice were fasted for 2 h, and then one cohort was subjected to WP while the other cohort was subjected to the control training paradigm (i.e., they experienced the same number of touch/shock cycles as the mice that were subjected to WP, but they only pulled an unweighted cart). At the end of this training session, the mice were returned to a cage that contained water but no food. At 30 min after completing the training session, the mice were given an intraperitoneal (IP) injection of 0.04 μmol/g bodyweight of puromycin (MilliporeSigma, Burlington, MA, USA) to measure the rate of protein synthesis as previously described [29]. At exactly 30 min after the injection of puromycin (i.e., 1 h post-training), the FDL muscle from the left leg of each mouse was collected, immediately frozen in liquid nitrogen, and stored at −80 °C. Of note, this experimental approach was used because it helped to minimize the potentially confounding effects of stress that occur during the initial WP sessions (e.g., stress that is due to a lack of experience with the handling procedures that occur during WP, as well as the damage/stress that occurs when naive muscles are subjected to a bout of intense exercise). Most importantly, the only difference between the WP and control group was whether the mice pulled a weighted vs. unweighted cart during the final training session (i.e., whether they performed the WP or control training paradigm). Thus, our experimental design allowed for a clear interpretation of the acute responses that are elicited by WP, rather than the responses that could have accumulated throughout several control vs. WP training sessions.

### 2.9. Western Blot Analysis

Frozen muscles were homogenized in ice-cold buffer B [40 mM Tris (pH 7.5), 1 mM EDTA, 5 mM EGTA, 0.5% Triton X-100, 25 mM β-glycerolphosphate, 25 mM NaF, 1 mM Na3VO4, 10 μg/mL leupeptin, and 1 mM PMSF] with a Polytron (PT 1200 E, Kinematica, Luceme, Switzerland), and then the homogenates were centrifuged at 6000 g for 1 min to remove bubbles and confirm complete homogenization. The entire sample was then thoroughly vortexed to resuspend the insoluble material, and an aliquot of the whole homogenate was subjected to further analysis. Specifically, the concentration of protein in the whole homogenates was determined with the DC protein assay kit (Bio-Rad, Hercules, CA, USA). Equal amounts of protein from each sample were dissolved in Laemmli buffer, subjected to SDS-PAGE, and transferred to a PVDF membrane as previously described [30]. At this point, the total protein levels on the membranes that were destined for the analysis of puromycin-labeled peptides (i.e., protein synthesis) were detected with “No-Stain” protein labeling reagent, as detailed in the manufacturer’s protocol (Invitrogen, Waltman, MA, USA). All membranes were then blocked with 5% milk-TBST and incubated with primary and secondary antibodies as previously described [30]. All blots were imaged with a UVP Autochemi system (UVP, Upland, CA, USA) along with the use of either regular enhanced chemiluminescence (ECL) reagent (Pierce, Rockford, IL) or ECL-prime (Amersham, Piscataway, NJ, USA). After imaging, some blots were incubated in stripping buffer [100 mM B-Mercaptoethanol, 2% SDS, and 62.5 mM Tris (pH 6.8)] for 15 min at 50 °C. These membranes were then blocked with 5% milk-TBST, incubated with new primary and secondary antibodies, and imaged as described above. In all cases, once appropriate images had been captured, Coomassie Blue staining was performed on the PVDF membrane and used to verify equal protein loading/transfer across the membranes. Images were quantified with ImageJ (https://imagej.nih.gov/nih-image/, accessed on 16 August 2021).

### 2.10. Western Blot Antibodies

Antibodies targeting 4E-BP1 (1:1000, #9644), eEF2 (1:3000, #2332), phosphorylated eEF2(56) (1:1000, #2331), eIF4B (1:1000, #3592), phosphorylated eIF4B (422) (1:1000, #3591), MKK4 (1:1000, #3346), phosphorylated MKK4 (257) (1:1000, #4514), p38 (1:1000, #9212), phosphorylated p38 (180/182) (1:1000, #9211), p70S6K (1:1000, #9234), phosphorylated p70S6K (389) (1:1000, #2708), S6 (1:1000, #2217), and phosphorylated S6 (240/244) (1:1000, #5364) were purchased from Cell Signaling Technology (Danvers, MA, USA). Anti-puromycin (1:5000, 12D10) was purchased from MilliporeSigma (Burlington, MA, USA). Peroxidase-labeled anti-mouse IgG2a (1:20,000, 115-035-206) was obtained from Jackson ImmunoResearch Laboratories, Inc (West Grove, PA, USA). Peroxidase-labeled anti-rabbit (1:5000, PI-1000) was obtained from Vector Labs Inc. (Burlingame, CA, USA).

### 2.11. Statistical Analyses

Statistical significance was determined by using the unpaired *t*-tests, one-way ANOVA, repeated measures one-way ANOVA, two-way ANOVA, or repeated measures two-way ANOVA. For all ANOVA-based analyses, multiple post hoc comparisons were FDR corrected with the two-stage step-up method of Benjamini, Krieger, and Yekutieli. Differences between groups were considered significant when *p* ≤ 0.05, or *q* ≤ 0.05 (i.e., when an FDR correction was employed). Prior to performing statistical analyses, the datasets were searched for outliers and individual data points that resided more than 3 standard deviations away from the mean were removed. All statistical analyses were then performed by GraphPad Prism 9, and the type of statistical analysis that was performed for each experiment is described in the figure legends. A detailed summary of the data and all of the associated statistical outcomes can be found in the Appendix A.

## 3. Results

### 3.1. Components of the Weight Pulling Model

A description of the custom-built components that were used for the weight pulling model is provided in Figure 1. Most of these components were purchased from a local hardware store, their assembly only required basic tools (e.g., screwdriver, saw, drill, and staple gun), and the total cost was very affordable (<USD 150). Of these components, the grip support material highlighted in Figure 1D is considered to be of particular importance. Specifically, to develop a large amount of force during the weight pulling movement, the mice needed to establish a solid grip with the base of the weight pulling lane. This could have been accomplished by lining the base of the lane with fine materials, such as a thin wire mesh; however, such materials would have put the mice at risk for developing cuts/wounds on their paws. Thus, to address this, we tested a variety of different types of grip support material and were particularly satisfied with the Con-Tact Brand shelf liner model #05F-C6B0B-06. This fine, yet semi-elastic, material enabled the mice to establish a solid grip, but signs of injury to the paws were never observed. In fact, the only issue with the material was that the quality slowly degraded with repeated use. Accordingly, the material was replaced after every ~100 training sessions.

### 3.2. Weighted Pulling Is a Full-Body/Multi-Joint Exercise

To help the reader appreciate what it looked like when mice were subjected to weight pulling, we recorded regular- and slow-motion videos (Appendix A). As shown in these videos, mice that pulled an unweighted cart (control mice) used a movement pattern that resembled the act of running. On the other hand, mice that engaged in weighted pulling (WP) generally needed to perform 8–12 vigorous movements to pull the cart the length of the weight pulling lane (i.e., 50 cm). Each vigorous movement typically began with flexion of the shoulders and, in most cases, continued until the front paws fell in line with the ears. At this stage, the hindlimbs were typically in a neutral position and the ankles were highly dorsiflexed. The mice would then simultaneously engage in a pulling motion with the forelimbs and a pushing motion with the hindlimbs. Of note, the pushing of the hindlimbs was typically driven by a substantial plantar flexion of the ankles and the success of the plantar flexion appeared to require a large amount of force from the digits in the rear paws (Figure 1D). As forward progress was made, the front paws would eventually reach the level of the pectoral region and, at this stage, the forelimb action would transition from a pulling to pushing motion. The final act of pushing with the forelimbs would then continue until the front paws reached the superior region of the abdomen. Collectively, these motions indicate that WP is a full-body/multi-joint exercise that requires the use of numerous muscles throughout the body.

### 3.3. Basic Characteristics of the Training Paradigm

To determine whether WP could induce an increase in muscle mass, mice were subjected to 13 weeks of training with either the WP or control (unweighted) paradigm. Throughout this period, each mouse performed three training sessions per week with each training session consisting of 8.3 ± 1.6 sets per session, and 2 min of rest between each set (Figure 2A). Importantly, all WP sets that occurred after the sixth set were performed with a load that was ≥100% the animal’s previous maximal pulling load. In other words, during each training session, WP mice performed approximately two sets with a load that was at, or above, their previous maximum. Based on these points, it can be concluded that the basic characteristics of the WP paradigm are consistent with the ACSM recommendations for models of PRE in humans [6].

In addition to mimicking the basic characteristics of a human PRE training paradigm, WP also led to strength changes that mirror what occurs when humans engage in PRE. For instance, as shown in Figure 2B, WP led to a rapid increase in the maximal load that the mice could successfully pull (i.e., strength). The rapid increase in strength was maintained throughout the first 6 weeks of training, and then the rate of increase largely plateaued. Notably, the temporal nature of this response was remarkably similar to what occurs in humans (see Figure 11 in Sale 1988 [31]). Thus, as described by Sale 1988, we strongly suspect that the initial increase in strength was largely due to neural adaptations, whereas the later increases were mediated by muscular adaptions.

Another attribute of WP that is consistent with human models of PRE is that the final set during each session was taken to failure [32]. Specifically, failure was defined as the inability of the mouse to make forward progress after five consecutive touches of the rear fur followed by one shock and then two additional touches. These details bear mentioning because the number of touches and shocks that were delivered during each training session progressively decreased over time (Figure 2C,D). In fact, after 4 weeks of training, the average number of shocks remained at or below 1.2 per session, which suggests that a maximal effort from the mice could have been elicited with a less assertive motivation procedure (e.g., a puff of compressed air). Moreover, the average number of touches per session declined from a peak of 68 during the 2nd week of training to 10 during the last week of training. In other words, even though the final set of each training session was taken to failure, the act of WP became a largely conditioned/self-motivated exercise.

### 3.4. WP Leads to a Decrease in Fat and an Increase in the Mass of Numerous Skeletal Muscles

As illustrated in Figure 3A, mice subjected to 13 weeks of training gained an average of 1.3 ± 1.2 g of bodyweight, and the change in bodyweight was not influenced by the type of training that the mice performed. Upon completing the 13 weeks of training, EchoMRI was used to assess body composition, and the outcomes revealed that WP led to a significant reduction in body fat (Figure 3B). At this point, the mice were also assessed for rotarod performance and grip strength and, as shown in Figure 3C,D, the results indicated that WP did not alter rotarod performance, but it did lead to a substantial (42%) increase in isometric grip strength. After obtaining these measurements, the mice were subjected to a terminal collection procedure and the properties of numerous tissues were assessed. For instance, it was determined that WP did not alter the mass of lean tissues, such as the heart or the adrenal glands, but it did lead to a significant decrease in the mass of the epidydimal fat pads (Figure 3E–G). Images of skinned forelimbs were also collected, and these images enabled us to determine that WP led to a significant increase in the maximal diameter of both the upper arm and forearm (Figure 3H–J). Moreover, we collected a diverse array of skeletal muscles from different regions of the body, including the hindlimbs, forelimbs, and torso, and it was determined that WP led to a 6–23% increase in the mass of 10 of the 15 different muscles that we examined (Figure 3K). Thus, just like in human models of PRE, WP not only increases strength, but also leads to an increase in skeletal muscle mass and a decrease in body fat.

### 3.5. WP Induces Hypertrophy and Myonuclear Accretion in the Plantaris Muscle

Numerous studies have shown that, in humans, the PRE-induced increase in skeletal muscle mass is associated with an increase in the cross-sectional area (CSA) of muscle fibers (i.e., hypertrophy) [23,33]. To examine whether the same relationship is observed in muscles that have been subjected to WP, we first measured various structural properties of the plantaris muscle. We initially focused on the plantaris because this has been one of the most widely studied muscles in other mouse models of PRE, and, as shown in Figure 3K, WP led to a modest but significant increase in its mass [17,20,34]. Our structural analyses also revealed that 13 weeks of WP did not alter the resting length of the plantaris, but it did lead to a 21% increase in the mid-belly CSA (Figure 4A–C). Moreover, we determined that the increase in mid-belly CSA was associated with an 18% increase in the average fiber CSA, and that the increase in fiber CSA was not due to a change in the aspect ratio of the fibers (Figure 4D,E). Finally, our analyses also revealed that WP did not alter the number of fibers per mid-belly cross-section (Figure 4F). Hence, it appears that the WP-induced increase in the mass of the plantaris was largely driven by hypertrophy of the individual fibers.

Next, we wanted to determine whether WP led to fiber type-specific differences in the hypertrophic response. As shown in Figure 4G–L, our analyses revealed that the plantaris is primarily composed of Type IIA, IIX, and IIB fibers (only 0.3% of the fibers were Type I), and no fiber type-specific differences in the hypertrophic response were detected. Moreover, WP did not alter the fiber type composition of the plantaris muscle.

In addition to inducing hypertrophy, numerous human studies have shown that the PRE-induced hypertrophic response is associated with myonuclear accretion [26,35]. Therefore, to determine whether WP induces this effect, we measured the myonuclei to fiber ratio in the mid-belly cross-sections and found that WP increased this ratio. We also determined that the myonuclear domain (i.e., myonuclei/fiber CSA) was not altered by WP, whereas the interstitial nuclei to fiber CSA ratio was significantly reduced (Figure 4M–P). Our analyses also revealed that WP did not alter the number of centrally located nuclei per fiber (0.006 ± 0.005 vs. 0.008 ± 0.005 for CNT vs. WP, respectively, *p* = 0.32). Collectively, these results indicate that, just like in human models of PRE, WP-induced hypertrophy is associated with myonuclear accretion.

### 3.6. WP Leads to an Increase in the Resting Length of the Lateral Head of the Triceps Brachii

In addition to inducing hypertrophy, a number of human studies have suggested that PRE can induce longitudinal growth of the fibers (i.e., the in-series addition of sarcomeres) [24,36]. Hence, we were intrigued by the changes that were observed in the lateral head of the triceps brachii (Tri-Lat). Specifically, as shown in Figure 3K, the Tri-Lat revealed one of the most robust WP-induced increases in mass (17%) and, as shown in Figure 5, this increase in mass was associated with a 10% increase in the resting length of the muscle. Interestingly, the mid-belly CSA of the Tri-Lat was not significantly altered, and there was actually a trend for a decrease in the average CSA of the fibers (*p* = 0.07). However, although WP did not induce hypertrophy, it did lead to a 17% increase in the number of fibers per cross-section. This result is important to highlight because we recently described how an increase in fiber length can lead to an increase in the number of fibers per cross-section [23,37]. Thus, we strongly suspect that WP induced the longitudinal growth of the fibers but, unfortunately, the collection procedures employed in our study did not allow for us to directly answer the question. Moreover, it remains possible that the increase in the number of fibers per cross-section was not due to longitudinal growth, but was instead mediated by events such as myofiber splitting [38]. As such, further studies will be needed to address these possibilities.

### 3.7. WP Induces Hypertrophy, a Type IIB-to-IIX Fiber-Type Switch, and Myonuclear Accretion in the FDL

To further characterize the types of muscular adaptations that are induced by WP, we next focused on the flexor digitorum longus muscle (FDL). We focused on the FDL because our analysis of the movement patterns (Appendix A) indicated that WP requires a large amount of force to be generated from the digits in the rear paws. From an anatomical standpoint, the FDL would be the primary muscle that is used to generate these forces, and, therefore, we reasoned that it is highly recruited during WP. In support of this notion, we found that 13 weeks of WP led to an 11% increase in the mass of the FDL (Figure 3K). However, unlike the Tri-Lat, the resting length of the FDL was not significantly altered (Figure 6). Our analyses also identified a trend for an increase in the mid-belly CSA of the FDL (13%, *p* = 0.08), and this trend was associated with an 11% increase in the average fiber CSA (Figure 6C,D). Moreover, we determined that the increase in the average fiber CSA was not due to alterations in the aspect ratio of the fibers and that WP did not alter the number of fibers per mid-belly cross-section (Figure 6E,F). Hence, in contrast to the Tri-Lat, the WP-induced increase in the mass of the FDL appears to be driven by hypertrophy of the individual fibers.

We next wanted to assess whether WP resulted in any fiber type-specific differences in the hypertrophic response. As shown in Figure 6G,H, the results indicated that, just like the plantaris muscle, the FDL is largely composed of Type IIA, IIX, and IIB fibers (only 0.2% of the fibers were Type I), and no fiber type-specific differences in the hypertrophic response were detected. However, unlike the plantaris muscle, WP led to a substantial decrease in the number of Type IIB fibers and an increase in the number of Type IIX fibers.

Previous human studies have shown that PRE can induce a fast-to-slow shift in the Type II fiber composition, and, thus, the WP-induced IIB-to-IIX fiber-type switch was not surprising [25,39,40]. However, the impact that this alteration could have on the average size of the individual fiber types was completely unforeseen. For instance, as shown in Figure 6G, the WP-induced increase in the CSA of the Type I, IIA, IIX, and IIB fibers was 21, 27, 37, and 18%, respectively. However, when all of the fibers were considered as a single group, the WP-induced increase in the CSA was only 11% (Figure 6D). The math simply did not add up, and, at first glance, the reason for the discrepancy was difficult to reconcile. However, a major strength of using mice, along with the analytical procedures that we employed, is that we were able to quantify the size and type of every fiber in the mid-belly cross-sections of the muscles, and thus, we were able to create frequency histograms that display the total number of fibers per fiber CSA (Figure 6I–L). This is different from the more commonly used approach of generating frequency histograms that display the percentage of the population per fiber CSA (Figure 6M–P), and it had a notable impact on our ability to interpret the results. For instance, the frequency histogram in Figure 6P suggests that the entire population of Type IIB fibers underwent a rightward shift (i.e., a shift towards a larger CSA). However, when the data are shown on a frequency histogram that displays the total number of fibers per fiber CSA, it appears that WP did not alter the highest (right) tertile of the Type IIB population (Figure 6L). Instead, the outcomes suggest that WP led to a preferential decrease in the number of fibers that belonged to the middle and lowest tertiles of the population, and we suspect that it was this particular portion of the population that switched to Type IIX fibers.

In mouse skeletal muscles, there is a well-recognized fiber type-specific difference in CSA with Type IIB > IIX > IIA ~ I [41]. Consistent with this point, the average CSA of Type IIB fibers in the control group was 1356 µm^2^, whereas the average for the Type IIX fibers was only 868 µm^2^. This is important because, if WP simply led to a Type IIB-to-IIX conversion (no WP-induced increase in CSA), then the magnitude of the WP-induced fiber type conversion alone would have brought the average CSA of the Type IIX fibers in the control group up to 1070 µm^2^. In other words, our results suggest that the WP-induced Type IIB-to-IIX conversion created the illusion of a ~23% WP-induced increase in the CSA of the Type IIX fibers. As mentioned above, the actual WP-induced increase in the CSA of the Type IIX fibers was 37%. Based on the points raised above, we strongly suspect that this was due to a combined effect of the WP-induced Type IIB-to-IIX conversion and a true WP-induced hypertrophy of the Type IIX fibers. The same general principles also likely contributed to the increase in the CSA that was observed in the other fiber types.

To complete our characterization of the adaptations that were induced by WP in the FDL, we next measured the myonuclei to fiber ratio in the mid-belly cross-sections. Similar to what was observed in the plantaris muscle, the outcomes revealed that WP led to an increase in this ratio, no change in the myonuclear domain, and a reduction in the interstitial nuclei to fiber CSA ratio (Figure 6Q–T). Moreover, we determined that WP did not alter the number of centrally located nuclei per fiber (0.009 ± 0.004 vs. 0.012 ± 0.005 for CNT vs. WP, respectively, *p* = 0.10).

### 3.8. WP Induces an Increase the Rate of Protein Synthesis and Alters the Phosphorylation State of Proteins That Are Regulated by PRE in Humans

Having established that WP can induce the same type of long-term adaptations that occur in human models of PRE, we next wanted to determine whether WP could induce some of the acute types of responses that are known to occur after a bout of PRE. For instance, in humans, it has been shown that PRE induces an acute: (i) increase in the phosphorylation state of various mitogen-activated protein kinases, (ii) activation of signaling by growth regulatory molecules, such as mTORC1, and (iii) increase in the rate of protein synthesis [21,42,43]. Thus, to determine whether WP could elicit these types of responses, mice were subjected to the control or WP conditions described in Figure 7A. As shown in Figure 7B,C, the outcomes revealed that, at 1 h after the bout of training, WP had led to an increase in the phosphorylation state of mitogen-activated protein kinases, such as p38 and MKK4. WP also induced signaling through various markers of mTORC1 activity, including an increase in p70S6K (389) and S6 (240/4) phosphorylation, and an increase in γ:total ratio of 4EBP1 (Figure 7D–F) [30]. Moreover, WP led to an increase in the amount of phosphorylated and total eIF4B, as well as a decrease in the amount of phosphorylated eEF2 and an increase in total eEF2 (Figure 7G,H). These later changes are particularly interesting because they suggest that WP induced an increase in the rate of translation initiation and elongation, respectively [44,45]. Consistent with this point, our analyses also revealed that WP led to an increase in the amount of puromycin-labeled peptides (i.e., the rate of protein synthesis) (Figure 7I). Collectively, these results indicate that WP can induce the same types of acute responses that are known to occur after humans engage in PRE.

## 4. Discussion

As stated in the introduction, one of the primary goals of this study was to develop a mouse model that mimics a human PRE training paradigm and induces an increase in the mass of numerous muscles throughout the body. As detailed in the results, the basic characteristics of the WP training paradigm were well aligned with the ACSM’s recommendation for models of PRE that are aimed at inducing muscular growth in humans (i.e., major muscle groups should be trained two to three times per week with an emphasis placed on multi-joint exercises that involve 8–12 repetitions per set, 1–2 min of rest between sets, and a total of one to three maximal-intensity sets of each exercise per training session). Moreover, the training paradigm fulfilled the ACSM’s recommendation of progressively increasing resistance as the individual’s strength improved [6]. Hence, our goal of developing a mouse model that mimics a human PRE training paradigm was accomplished.

The results presented in Figure 3 indicate that we also accomplished our goal of developing a mouse model that can induce an increase in the mass of numerous muscles throughout the body. Indeed, we observed a 6–23% increase in the mass of 10 of the 15 different muscles that were examined, which is very similar to the 5–20% increase in mass/volume that has been reported in humans after 8–16 weeks of PRE [8,9,10]. We also determined that our model can be used to gain insight into the long-term adaptations that lead to the increases in mass. For instance, we demonstrated that WP can be used to study the mechanisms via which PRE induces hypertrophy, myonuclear accretion, and fiber type switching (Figure 4 and Figure 6). Moreover, the results in Figure 5 suggest that WP could also potentially be used to study how PRE induces the longitudinal growth of fibers.

WP not only induced an increase in the mass of numerous skeletal muscles, but it also led to whole-body level changes that are commonly observed with PRE. For instance, in humans, it has been shown that PRE can lead to a decrease in body fat percentage [11,46,47]. Consistent with these reports, we found that WP led to a 26% reduction in the mass of the epidydimal fats pads and a 22% reduction in the percentage of the total body mass that was composed of fat (Figure 3B,G). This is noteworthy because the induction of whole-body level changes suggests that WP could serve as an effective model for defining the mechanisms via which PRE induces system-wide improvements in metabolic and cardiovascular health, and how PRE can act as an effective countermeasure against aging-related diseases, such as sarcopenia and cancer [11,12,13,14].

Another major strength of the weight pulling model is that, aside from being capable of inducing the classic long-term adaptations that occur in humans (e.g., hypertrophy, myonuclear accretion), it also recapitulates the acute responses that are thought to drive these adaptations. For instance, current dogma asserts that PRE induces the activation of mTORC1-dependent signaling events, and that these events, in turn, promote an increase in the rate of protein synthesis and the concomitant increase in muscle mass [48,49,50]. However, support for this paradigm has largely come from studies that used the drug rapamycin to inhibit mTORC1 signaling and/or a model of chronic mechanical overload (e.g., synergist ablation) to induce an increase in muscle mass [22,30,51,52,53]. With the advent of the weight pulling model, it will now be possible to test this paradigm within the confines of a physiologically relevant stimulus. Moreover, the weight pulling model will allow for the use of established genetic inventions that are often paramount to mechanistic studies. As a case in point, we previously developed mice with a skeletal muscle-specific and inducible knockout of Raptor/mTORC1 [30], and these mice could be used to determine whether signaling through mTORC1 in the skeletal muscle is necessary for the WP-induced increase in protein synthesis, muscle mass, and strength, as well as system-wide changes, such as the reduction in body fat, etc.

Like all models, the weight pulling model has unique strengths, but it also has limitations. In our opinion, the greatest limitation is that, unlike other recently described mouse models of PRE, the weight pulling model requires a significant amount of hands-on time from investigators (~20 h per week are required for an investigator to train 10 control and 10 WP mice) [16,34]. However, it is important to consider that approximately the same amount of time would be required for an investigator to train an equal number of human subjects (personal communication from Dr. Stuart Phillips, McMaster University). Moreover, we suspect that the bulk of the long-term adaptations reported in this study would have been detected after just 6 weeks of training, rather than the 13 weeks that were employed. The basis for this argument stems from the data in Figure 2B, which reveal that the gains in strength substantially plateaued after the 6th week of training (i.e., 90% of the total increase in maximal pulling load occurred during the first 6 weeks of training). According to Sale 1998, muscle and strength adaptations plateau at the same time after the onset of PRE [31]. If this point holds true in the weight pulling model, then transitioning from 6–13 weeks of training would have only led to a ~10% greater increase in variables such as muscle mass (e.g., a 10% increase in mass after 6 weeks vs. an 11% increase after 13 weeks). Thus, we strongly suspect that the long-term adaptations observed in this study could have been achieved with as little as 120 h of investigator time. However, even if the full 13 weeks of training is necessary, the translatable nature of the outcomes could still be well worth the time that is required in order to obtain them.

## 5. Conclusions

The results of this study describe a cost-effective mouse model of PRE that is based on a full-body/multi-joint exercise along with a training paradigm that mimics human PRE (three training sessions per week, 8–12 repetitions per set, 2 min of rest between sets, approximately two maximal-intensity sets per session, last set taken to failure, and progressive increase in loading that is based on the individual’s performance). We demonstrate that WP is capable of inducing an increase in the mass of numerous muscles throughout the body, and that the increase in mass is associated with the same type of long-term adaptations that are known to occur in humans (fiber hypertrophy, myonuclear accretion, and, in some instances, a fast-to-slow transition in the composition of the Type II fibers). Moreover, we demonstrate that WP can induce the same type of acute responses that are thought to drive these long-term adaptations (e.g., activation of signaling through mTORC1 and the induction of protein synthesis). When taken together, the results of our study suggest that WP is a highly translatable mouse model of human PRE. Hence, we propose that our model will not only help investigators to gain better insight into the mechanisms via which PRE induces an increase in muscle mass, but will also open the door for new types of studies that are aimed at determining how PRE leads to improvements in overall health and disease prevention.

## Figures and Tables

**Figure 1 cells-10-02459-f001:**
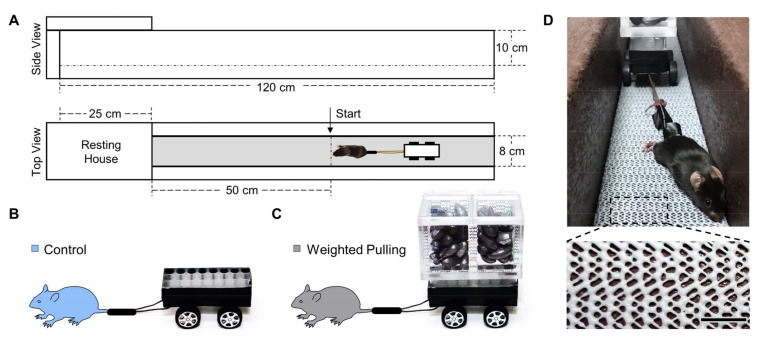
Weight pulling model components: (**A**) schematic of the weight pulling track and the associated dimensions of the primary components; (**B**,**C**) images of the unweighted and weighted versions of the weight pulling cart, respectively; (**D**) image of a mouse pulling a weighted cart. Higher magnification of the boxed region reveals the texture of the grip support material that was used to line the base of the weight pulling track. Scale bar = 1 cm.

**Figure 2 cells-10-02459-f002:**
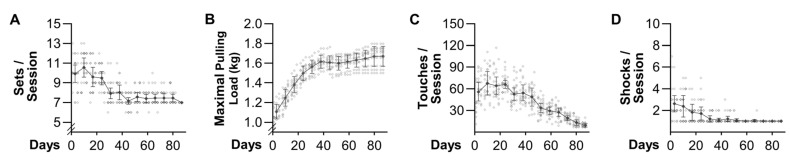
Basic characteristics of the training paradigm. Mice were subjected to 13 weeks of weighted pulling (WP) and, during each training session, various parameters were measured including: (**A**) the number of sets that were successfully completed; (**B**) the maximal load that was successfully pulled the entire length of the lane; (**C**) the number of times the rear was touched; and (**D**) the number of shocks that were delivered. Individual data points are indicated with hollow symbols and weekly means for the entire group are indicated with solid symbols. The group data are presented as the mean ± SD (*n* = 10/group). All datasets were analyzed with repeated measures one-way ANOVA. Multiple pairwise comparisons were FDR corrected with the two-stage step-up method of Benjamini, Krieger, and Yekutieli and the detailed outcomes of these analyses can be found in the Appendix A.

**Figure 3 cells-10-02459-f003:**
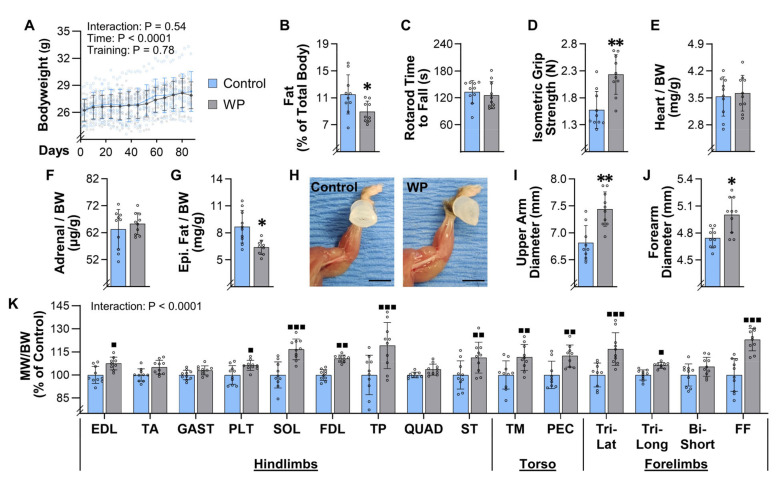
Weight pulling leads to a decrease in fat and an increase in the mass of numerous skeletal muscles. Mice were subjected to 13 weeks of training with weighted pulling (WP) or the unweighted (control) paradigm: (**A**) bodyweight (BW) was recorded at the start of every training bout (individual data points are displayed with hollow symbols and the weekly means for each group are displayed with solid symbols); (**B**–**K**) after 13 weeks of training, the mice were subjected to measurements of (**B**) body fat composition, (**C**) rotarod performance, and (**D**) maximal isometric grip strength; (**E**) the weight of the heart, and the average weight of (**F**) the left and right adrenal glands and (**G**) epididymal fat pads, were measured and expressed relative to BW; (**H**) images of the left forelimb were taken and used to measure the (**I**) maximal upper arm diameter, and (**J**) maximal forearm diameter; (**K**) numerous skeletal muscles from the hindlimbs, torso, and forelimbs were also collected, and then the average muscle weight (MW) to BW ratio for each pair of muscles (i.e., from the left and right side of the body) was determined and expressed relative to the mean value obtained in the control group. Bar graphs represent the group means ± SD, with individual data displayed as hollow circles (*n* = 9–10/group). The results in A were analyzed with repeated measures two-way ANOVA, B-G and I-J were analyzed with unpaired *t*-tests, and K was analyzed with two-way ANOVA. For A and K, multiple post hoc comparisons were FDR corrected with the two-stage step-up method of Benjamini, Krieger, and Yekutieli. The insets in A and K indicate the *p*-value for the main effects and/or interaction. Significantly different from condition-matched control group, * *p* < 0.05, ** *p* < 0.005, ■ *q* < 0.05, ■■ *q* < 0.005, ■■■ *q* < 0.0005. Scale bar in H = 5 mm. Note: the results from all of the multiple comparisons in A can be found in the Appendix A.

**Figure 4 cells-10-02459-f004:**
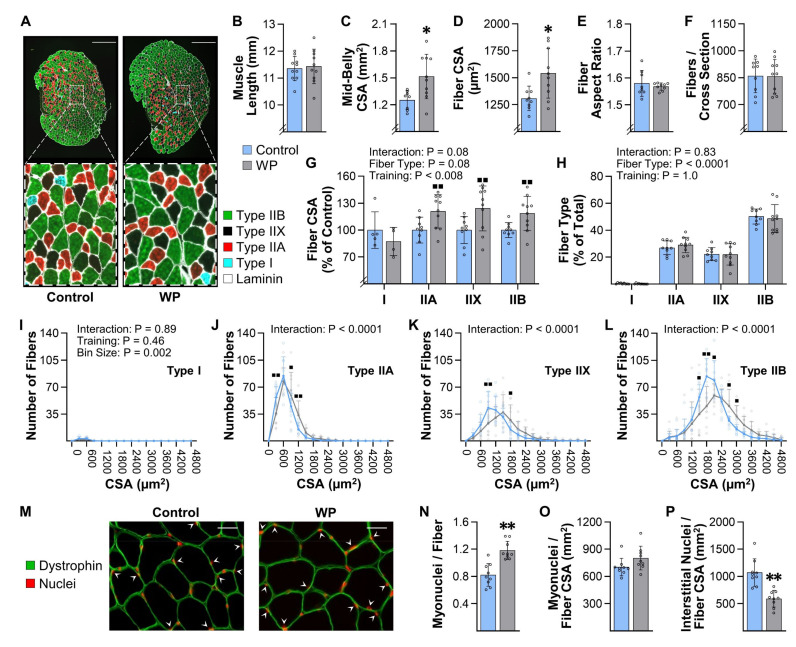
WP induces hypertrophy and myonuclear accretion in the plantaris. Plantaris muscles were collected from mice that had completed 13 weeks of training with weighted pulling (WP) or the unweighted (control) paradigm: (**A**) representative mid-belly cross-sections from samples that had been subjected to immunohistochemistry (IHC) for laminin and fiber type identification (i.e., Type I, IIA, IIX, or IIB); (**B**) prior to the IHC analyses, the resting length of the plantaris muscles from both the right and left leg of each animal was measured, and then the average for the pair was recorded; (**C**–**L**) one muscle from each animal was subjected to IHC as described in (**A**), and then the entire cross-section was analyzed to determine the (**C**) mid-belly cross-sectional area (CSA), (**D**) average CSA of the fibers, (**E**) average aspect ratio (i.e., max ferret diameter/min ferret diameter) of the fibers, and (**F**) the total number of fibers in the cross-section. The same images were also used to determine the effect of WP on the (**G**) relative CSA of the individual fiber types, and (**H**) proportion of the fibers that were represented by the different fiber types; (**I**–**L**) frequency histograms that display the number of fibers from each muscle that had the indicated fiber CSA; (**M**) representative mid-belly cross-sections from plantaris muscles that were subjected to IHC for dystrophin (to identify the outer boundary of the muscle fibers) and Hoechst (to identify nuclei). Images of the entire cross-section were analyzed to determine the (**N**) myonuclei to fiber ratio (i.e., the total number of nuclei within the muscle fibers/total number of muscle fibers), (**O**) the myonuclei to fiber CSA ratio (i.e., the total number of myonuclei/total CSA occupied by the muscle fibers), and (**P**) the interstitial nuclei to fiber CSA ratio (i.e., the total number of interstitial nuclei/total CSA occupied by the muscle fibers). All data are presented as the group mean ± SD, with individual data points displayed as hollow symbols (*n* = 9–10/group, except for the Type I fibers in G, which only had detectable Type I fibers in four to five of the muscles/group). The results in B–F and N–P were analyzed with unpaired *t*-tests, and the results in G–L were analyzed with two-way ANOVA followed by multiple post hoc comparisons that were FDR corrected with the two-stage step-up method of Benjamini, Krieger, and Yekutieli. The insets in G–L indicate the *p*-value for the main effects and/or interaction. Significantly different from the condition-matched control group, * *p* < 0.05, ** *p* < 0.0005, ■ *q* < 0.05, ■■ *q* < 0.005. Scale bar in A = 500 µm, scale bar in M = 25 µm, arrows in M point to myonuclei.

**Figure 5 cells-10-02459-f005:**
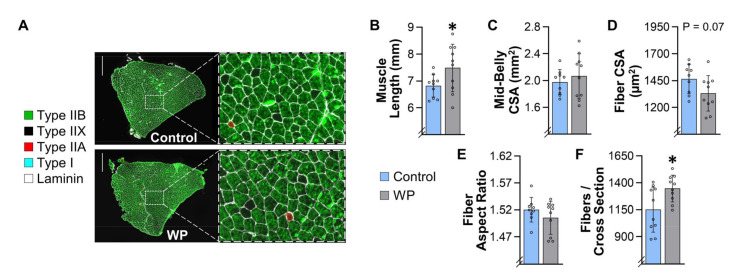
WP leads to an increase in the resting length of the lateral head of the triceps brachii (Tri-Lat). Tri-Lat muscles were collected from mice that had completed 13 weeks of training with weighted pulling (WP) or the unweighted (control) paradigm: (**A**) representative mid-belly cross-sections from samples that had been subjected to immunohistochemistry (IHC) for laminin and fiber type identification (i.e., Type I, IIA, IIX, or IIB); (**B**) prior to the IHC analyses, the resting length of the Tri-Lat muscles from both the right and left leg of each animal were measured, and then the average for the pair was recorded; (**C**–**F**) one muscle from each animal was then subjected to IHC as described in (**A**), and the entire cross-section was analyzed to determine the (**C**) mid-belly cross-sectional area (CSA), (**D**) average CSA of the fibers, (**E**) average aspect ratio (i.e., max ferret diameter/min ferret diameter) of the fibers, and (**F**) the total number of fibers in the cross-section. The data are presented as the group mean ± SD, with individual data points displayed as hollow symbols (*n* = 9–10/group). The results in B–F were analyzed with unpaired *t*-tests. Significantly different from the condition-matched control group, * *p* < 0.05. Scale bar in A = 500 µm.

**Figure 6 cells-10-02459-f006:**
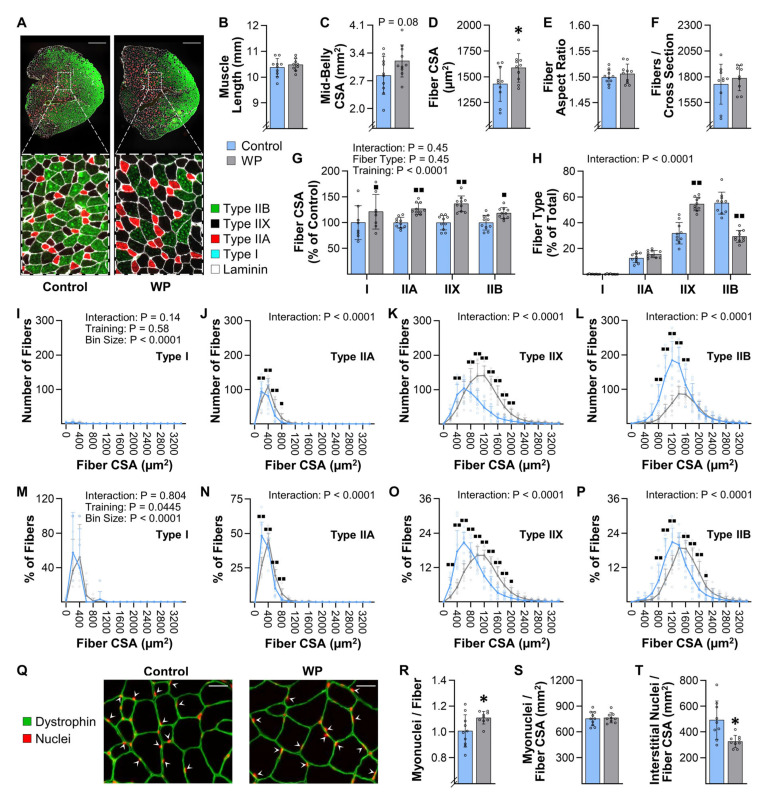
WP induces hypertrophy and Type IIB-to-IIX fiber-type switching in the FDL. FDL muscles were collected from mice that had completed 13 weeks of training with weighted pulling (WP) or the unweighted (control) paradigm: (**A**) representative mid-belly cross-sections from samples that had been subjected to immunohistochemistry (IHC) for laminin and fiber type identification (i.e., Type I, IIA, IIX, or IIB); (**B**) prior to the IHC analyses, the resting length of the FDL muscles from both the right and left leg of each animal were measured, and then the average for the pair was recorded; (**C**–**L**) one muscle from each animal was subjected to IHC as described in (**A**), and then the entire cross-section was analyzed to determine the (**C**) mid-belly cross-sectional area (CSA), (**D**) average CSA of the fibers, (**E**) average aspect ratio (i.e., max ferret diameter/min ferret diameter) of the fibers, and (**F**) the total number of fibers in the cross-section. The same images were also used to determine the (**G**) average CSA of the individual fiber types and (**H**) proportion of the fibers that were represented by the different fiber types; (**I**–**P**) frequency histograms that display the number of fibers within each muscle (**I**–**L**) or the percentage of the fibers within each muscle (**M**–**P**) that had the indicated fiber CSA; (**Q**) representative mid-belly cross-sections from samples that were subjected to IHC for dystrophin (to identify the outer boundary of the muscle fibers) and Hoechst (to identify nuclei). Images of the entire cross-section were analyzed to determine the (**R**) myonuclei to fiber ratio (i.e., the total number of nuclei within the muscle fibers/total number of muscle fibers), (**S**) the myonuclei to fiber CSA ratio (i.e., the total number of myonuclei/total CSA occupied by the muscle fibers), and (**T**) the interstitial nuclei to fiber CSA ratio (i.e., the total number of interstitial nuclei/total CSA occupied by the muscle fibers). The data are presented as the group mean ± SD, with individual data points displayed as hollow symbols (*n* = 8–10/group). The results in B–F and R–T were analyzed with unpaired *t*-tests, whereas the results in G–P were analyzed with two-way ANOVA followed by multiple post hoc comparisons that were FDR corrected with the two-stage step-up method of Benjamini, Krieger, and Yekutieli. The insets in G–P indicate the *p*-value for the main effects and/or interaction. Significantly different from the condition-matched control group, * *p* < 0.05, ■ *q* < 0.05, ■■ *q* < 0.01. Scale bar in A = 500 µm, scale bar in Q = 25 µm, arrows in Q point to myonuclei.

**Figure 7 cells-10-02459-f007:**
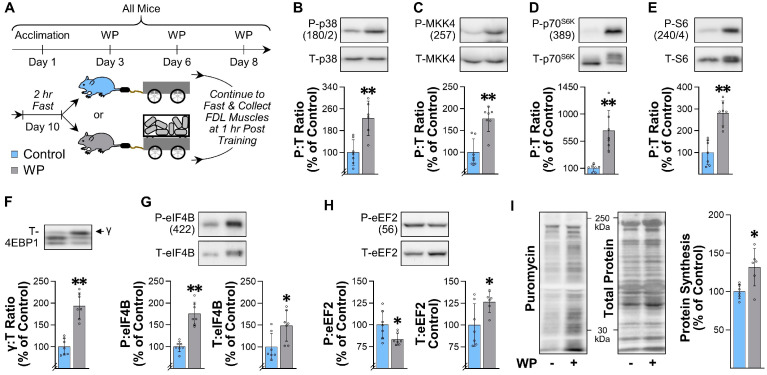
WP induces an increase the rate of protein synthesis and alters the phosphorylation state of proteins that are regulated by PRE in humans: (**A**) overview of the experimental design (see Section 2.8 for additional details). At 30 min after the final training session, the mice were injected with puromycin to label newly synthesized peptides, and then the FDL muscles were collected at 1 h post-training and subjected to Western blot analysis; (**B**–**H**) representative Western blots of the total (T) and phosphorylated (P) levels of the indicated proteins; (**I**) Western blots of puromycin-labeled peptides were normalized to the total protein levels and used to calculate the relative rate of protein synthesis. All values in the graphs are expressed relative to the mean value obtained in the control group, with bars representing the group mean ± SD, and individual data points displayed as hollow symbols (*n* = 6–7/group). All results were analyzed with unpaired *t*-tests. Significantly different from the control group, * *p* < 0.05, ** *p* < 0.005.

## Data Availability

All data needed to evaluate the conclusions or reperform analyses in the paper are presented in the manuscript and/or the Appendix A.

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
