# Peer review of "Weight Pulling: A Novel Mouse Model of Human Progressive Resistance Exercise"

_cells, 2021, doi:10.3390/cells10092459_

Round 1

Reviewer 1 Report

This is an interesting paper, which   describes an innovative method for whole body exercise  "Weight Pulling" -involving multiple body muscles in mice,  as opposed to traditional methods that often use Tread mill exercise. The design and execution of the methods appear to be well executed . The manuscript is well written and the data are clearly presented  . In fact,  it does present convincing data to suggests that this method is quite useful for inducing muscle growth/hypertrophy  and should be adapted for resistance training in mouse models which are very difficult to perform in mice, therefore  studies rely on treadmill exercise.   A potential limitation is getting  mice do the pulling without some incentives and training them which might be bit tricky . Other than that it can be adapted for certain studies where resistant exercise is desired  as a modality of treatment to prevent muscle loss. 

One concern I have is , in the absence of a standardized WP method, the investigators may have to rely on their own innovation , therefore it is necessary to commercialize this technique to  get  uniform results.

Reviewer 2 Report

Zhu et al. have assessed the effect of newly developed mouse model of progressive resistance exercise (PRE), namely weight pulling (WP), that mimics a traditional human paradigm. WP induces an increase in the muscle mass throughout the body. The WP-induced physiological adaptaions as well as molucular responses were similar to what has been observed in human, suggesting that WP is a highly translatable mouse model of human PRE. Although the experiments are well conducted and the manuscript is of interest, several concerns have been raised which are detailed below.

Major comments:

  1. Line 51: Synergist ablation, a surgical model of overload, has long been used to induce muscle hypertrophy in rodents, but this is regarded as supraphysiological overload model which induces muscle dysfunction and damage (Kandarian & Williams, Med Sci Sports Exerc 25: 999-1004, 1993). The rationale behind the present study would be strengthened by emphasizing the lack of translatability of this classical overload model in the Introduction section.
  2. Line 56: Figure 3K show that the magnitude of the increase in muscle mass differs markedly between the muscles. There are some translational models of muscle hypertrophy that more closely reflect human PRE (Murach et al., 2020). Among them, ”progressive weighted wheell running (Dungan et al., 2019; Murach et al., 2019)” and ”high-intensity incline treadmill running (Seldeen et al., 2018; Goh et al., 2019)” are similar to WP in terms of the exercise mode that involves walking and/or runnning. It means that the muscles recruited during exercise wolud be similar between these models. Although, as suggested by the authors, most models only focus on a couple of muscles within a single limb, it would be important to compare the magnitude of the increase in muscle mass in each muscles between the previous studies and the present study to see if there is any similalities or differences between the models. These should be discussed in the Discussion section.
  3. Moreover, it has been suggested that above mentioned two models induce muscle hypertrophy, but should be used with the understanding that training volume is high and that the stimulus is a combination of resistance and endurance. In this regard, the WP model would have an advantageous, because it mimics a traditional human paradigm of PRE. These points should be discussed in the Discussion to strengthen the novelty of the WP model.
  4. Line 518: The resting length of the muscle was increased by 10% in lateral head of the triceps brachii (Tri-Lat), but not in other muscles examined (i.e., plantaris and FDL). Please comment on the reason for these heterogeous adaptations.
  5. Line 523: I understand the point that subtle changes in fiber length could lead to a substantial increase in both the number of fibers per cross-section and the overall cross-sectional area of the muscle, how can you exclude the possibility that muscle fiber splitting is involved in the increased total number of fibers per cross-section in the Tri-Lat muscle (Murach et al., Exerc Sport Sci Rev 47: 108-115, 2019) ?

Miner comments:

Line 533: ’C-L’ should be ’C-F’.
